Developing a semi-pelagic trawl to capture redfish in the Gulf of St. Lawrence, Canada

Nguyen Vang Y. Vang.Nguyen@mi.mun.ca
Bayse Shannon M.
Winger Paul D.
DeLouche Harold
Legge George
Centre for Sustainable Aquatic Resources, Fisheries and Marine Institute, Memorial University of Newfoundland , St. John’s , Newfoundland and Labrador , Canada
Yapıcı Sercan
Electronic publication date: 2023 Oct 16
Publication date: 2023
Volume: 11
Electronic Location ID: e16244
Received 2023 Jun 26; Accepted 2023 Sep 14
Copyright: ©2023 Nguyen et al.
Copyright year: 2023
Copyright holder: Nguyen et al.
License: This is an open access article distributed under the terms of the Creative Commons Attribution License, which permits unrestricted use, distribution, reproduction and adaptation in any medium and for any purpose provided that it is properly attributed. For attribution, the original author(s), title, publication source (PeerJ) and either DOI or URL of the article must be cited.
License URL: https://creativecommons.org/licenses/by/4.0/

Keywords: Fish behaviour, Capture efficiency, Redfish, Semi-pelagic trawl, Redfish fishery, Gear modification, Trawl designs, Bycatch reduction, Groundgear, Seabed impacts

Funding: Canadian Centre for Fisheries Innovation (CCFI) Government of Newfoundland and Labrador through the Department of Industry, Energy and Technology The Canada First Research Excellence Fund (CFREF) through the Ocean Frontier Institute (Module H) Natural Sciences and Engineering Research Council of Canada (NSERC) This work was supported by the Canadian Centre for Fisheries Innovation (CCFI), the Government of Newfoundland and Labrador through the Department of Industry, Energy and Technology, The Canada First Research Excellence Fund (CFREF) through the Ocean Frontier Institute (Module H), and the Natural Sciences and Engineering Research Council of Canada (NSERC) Discovery Grant. The funders had no role in study design, data collection and analysis, decision to publish, or preparation of the manuscript.

==============================
In this study, we developed a semi-pelagic trawl to target redfish (Sebastes spp.) and potentially reduce the capture of bycatch species and seabed impacts in the Gulf of St. Lawrence, Canada. The new trawl used an innovative technique connecting the upper bridles of the trawl to the warps, anterior of the trawl doors, leading to the trawl system being fished off the seabed. Such a technique can be used to match the heights of redfish as they move above the seabed during their diurnal cycle while allowing bycatch species related to the seabed to escape under the trawl. A 1:10 scale model of the trawl was constructed and evaluated in a flume tank to optimize the rigging and then a full-scale trawl was constructed for sea trials. Two field experiments subsequently evaluated the trawl at sea. The first field experiment concentrated on the experimental trawl’s operation and video observations of redfish behaviour in the trawl mouth and its effect on trawl entry. The second field experiment concentrated on a small-scale preliminary test on the catch of redfish and bycatch species when the trawl was on or off the seabed. Capture results, though preliminary, indicate that redfish can be targeted commercially with a semi-pelagic trawl, though some redfish will escape under the trawl. Additionally, results suggest that the catches of bycatch species may be reduced. In conclusion, this study suggests that a semi-pelagic trawl could be considered an effective technique to harvest redfish sustainably.

Introduction

The redfish fishery in the Gulf of St. Lawrence (GSL) has been in moratorium since 1995 (Duplisea, 2018; Fisheries and Oceans Canada, 2020). Currently, only a small index fishery remains (2,000 t year −1). Catches in the redfish fishery are dominated by deepwater redfish (Sebastes mentella) and Acadian redfish (S. fasciatus), collectively called redfish (Fisheries and Oceans Canada, 2020). Redfish are slow-growing, late maturing, and long-living species (Cadigan et al., 2022) and thus can be susceptible to overfishing (Koslow et al., 2000). These characteristics combined with a high proportion of small fish (<22 cm (minimum landing size)) observed in the catch led to a moratorium in the GSL redfish fishery in 1995 (Duplisea, 2018). However, due to recent strong recruitment events, a large biomass of redfish is found in the GSL. This has led to an upcoming commercial redfish fishery in the GSL (Fisheries and Oceans Canada, 2020). The likelihood of an imminent fishery has led to the testing of conservation measures and management concepts in the attempt to maintain a sustainable fishery.

Several technical measures have been implemented to improve the size selectivity, i.e., to reduce the capture of undersize fish, of the groundfish trawl currently used to target redfish in the GSL (Fisheries and Oceans Canada, 2020). While the currently regulated codend (i.e., 90 mm diamond mesh codend) has poor size selectivity (Cheng et al., 2020), modifying the codend mesh construction can improve the size selectivity for redfish (Pol, 2016; Cheng et al., 2020). Prior experiments have shown that the capture of small roundfish could be reduced by using a codend with a different mesh configuration, where diamond mesh is rotated 90° in the transverse direction, called a T90 codend (Herrmann, Priour & Krag, 2007; Tokaç et al., 2014; Bayse et al., 2016). Cheng et al. (2020) applied T90 codends in the GSL redfish trawl fishery. These authors revealed that using T90 codends can reduce the capture of small redfish compared with the currently used 90 mm diamond mesh codend. Additionally, Nguyen et al. (2023) developed a shaking codend using the T90 codend that further improved the size selectivity of redfish. Increasing the mesh size of diamond mesh codends has also demonstrated potential for reducing the capture of small redfish (Pol, 2016).

Potential problematic catch rates of unwanted species are a concern once the GSL redfish fishery is reopened. This is considered a key issue in redeveloping a sustainable redfish fishery in the near future (Duplisea, 2018; Fisheries and Oceans Canada, 2021; Cadigan et al., 2022). Redfish co-occur with many other groundfish species in the Northwest Atlantic, including Atlantic cod (Gadus morhua), white hake (Urophycis tenuis), and flatfish (Gomes, Haedrich & Rice, 1992). These species are associated with the seabed, and thus, they have been captured as bycatch in the redfish trawl fishery (Fisheries and Oceans Canada, 2021). Atlantic cod and flatfish such as Greenland halibut (Reinhardtius hippoglossoides) and Atlantic halibut (Hippoglossus hippoglossus) could be captured along with redfish as they enter the trawl at heights close to the seabed (Ryer, 2008; Winger, Eayrs & Glass, 2010; Pol & Eayrs, 2021). The distribution of white hake has shifted to deeper waters, which now overlaps with redfish in the GSL (Miri & Simpson, 2006; Fisheries and Oceans Canada, 2021). The bycatch of Greenland halibut, white hake, and Atlantic cod made up 9% of the total landings of redfish fishery between 2000-2009 in the GSL (Fisheries and Oceans Canada, 2020). In the southern region of the GSL, a recent small-scale fishery and multi-species survey estimated white hake bycatch on average was 10.5% of the total catch, though the high average was likely due to large bycatch instances, the recorded median was between 0.0−1.6%, still white hake bycatch is a concern once the redfish fishery reopens (Fisheries and Oceans Canada, 2021). This diverse group of species represents different sizes and body shapes, which make the species selection process at the codend difficult.

The diurnal cycle of redfish has been observed where redfish are close to the seabed during the day and migrate up the water column at night (Beamish, 1966). As a semi-demersal species, redfish typically aggregate in small or large shoals, and make vertical movements from the seabed to intermediate layers, likely associated with feeding behaviour (Gauthier & Rose, 2005; Cadigan et al., 2022). This leads to redfish being distributed higher in the water column at night. Thus, redfish availability to be captured by bottom trawling is affected by the time of day (Atkinson, 1989; Casey & Myers, 1998; Gauthier & Rose, 2005). Given that redfish are commonly found off the seabed, an effective species-selective fishing approach could capture redfish at such heights, while also having the potential to reduce demersal species bycatch given the distance between redfish (in the water column) and bycatch (on the seabed).

Semi-pelagic trawling is designed as a hybrid technique that can capture fish distributed on- and off-seabed (He et al., 2021). Semi-pelagic trawling can be considered when doors are fished off-seabed and the trawl in on-seabed, trawl is off-seabed and the doors are on-seabed, or in a hybrid fishing situation where either doors or trawl are moved on- or off-seabed as fishing conditions or motivations change (He et al., 2021; Montgomerie, 2022). Such a rigging technique was developed in the 1990s to fish redfish in the GSL, commonly known as French rigging or fork rigging (Garner, 1978; He & Winger, 2010). This method maintains the doors on the seabed while raising the trawl net off the seabed by connecting the upper bridles to the warps anterior of the doors (Garner, 1978; He & Winger, 2010). In the GSL fishery, this technique was used to target redfish as they migrate off the sea floor and to avoid net damage from a rough seabed (L Dredge, pers. comm., 2021; Garner, 1978; He & Winger, 2010). However, this technical measure went into disuse since the redfish fishery was in moratorium in the GSL. Other fisheries have used a similar trawling technique in France and the United Kingdom targeting a variety of species (He & Winger, 2010).

Currently in the GSL, the Northern shrimp (Pandalus borealis) stock is at its historical lowest abundance (Fisheries and Oceans Canada, 2022). The major factors leading to this low stock level include deep water warming, low dissolved oxygen levels, and predation from redfish, none of which are expected to change soon (Fisheries and Oceans Canada, 2022). This scenario has led to a reduced total allowable catch (TAC) for Northern shrimp in the GSL and shrimp active licence holders (n = 114; Fisheries and Oceans Canada, 2022) are anxious to target redfish once the fishery reopens (The Ocean Frontier Institute Northwest Atlantic Redfish Symposium, 2018; Dean-Simmons, 2021). It could be beneficial if the shrimp fleet transitioned to target redfish. This transition would reduce the fishing pressure on the Northern shrimp stock and harvesting redfish would compensate a difficult economic situation for shrimp fishers. Thus, there is need to develop a trawl that can capture redfish off the seabed that would transition easily for the shrimp fleet.

Pelagic trawling is an effective method to target redfish off of the seabed (Duplisea, 2018), however for shrimp trawlers, a pelagic trawl would be excessively expensive (large trawl, new sensors, new doors, etc.) and/or require a more powerful vessel. A method such as semi-pelagic trawling (i.e., “French rigging”) could be a cost-effective solution for the shrimp fleet when targeting redfish off the seabed. This technique only requires a typical groundfish trawl and a relatively simple modification to the upper bridles and warps.

This study aimed to further develop a semi-pelagic trawl for the emerging redfish fishery in the GSL. The gear development applied the “French rigging” technique described above to modify the groundfish bottom trawl currently used for the redfish fishery. This technique focused on changes in connection between upper bridles and trawl doors to allow the trawl net to fish off the seabed while maintaining trawl doors on the seabed. By lifting the trawl net off the seabed, the semi-pelagic trawl could capture redfish as they migrate higher in the water column at night and potentially avoid the capture of main bycatch species related to the seabed (i.e., Atlantic cod, Atlantic halibut, white hake, etc.). The trawl’s performance and ability to capture redfish and bycatch species were described through flume tank testing and sea experiments, respectively. Redfish behaviour at the trawl mouth and its effect on trawl entry was also observed during sea experiments. A successful new trawl design would be a functional alternative to sustainably target redfish in the GSL and provide an economical alternative to target redfish off of the seabed for fishers transitioning from targeting Northern shrimp. Further, this study provides additional insights into redfish responses at the trawl entrance, which is not well understood.

Material and Methods

Model construction, flume tank test, and full-scale modifications

A 1:10 scale model of an existing groundfish balloon trawl was constructed using a combination of Froude and Newton scaling principles (Dickson, 1959; Fridman, 1973). Force and geometric modelling laws were used during the scaling process to approximate full-scale bottom trawl characteristics and performance (Araya-Schmidt et al., 2021) . The model was evaluated in a flume tank (Fig. 1) located at the Fisheries and Marine Institute of Memorial University of Newfoundland, Canada (Winger, Louche & Legge, 2006). The model was spread with a pair of Morgere PF doors and evaluated across a range of towing speeds and rigging configurations.

Figure 1 A 1:10 scale model was evaluated in the flume tank located at the Fisheries and Marine Institute of Memorial University of Newfoundland, Canada.

The full-scale net plan of the trawl is previously described in Cheng et al. (2020). It has a headline length of 40.2 m, and a fishing line of 44.5 m. The rockhopper groundgear consists of 40.6 cm ∅ rollers on average and the headline consists of 132 floats (20.3 cm ∅). The trawl belly sections were constructed with the same netting, 170 mm diamond PE twine with a 3.5 to 4.0 mm ∅ (Winger, Louche & Legge, 2006).

Off-seabed trawling is represented by seabed clearance, which was tested at different rigging scenarios. Seabed clearance is defined as the vertical distance between groundgear and the seabed (in this case the flume tank belt), calculated by subtracting headline height (the height of the headline to the seabed) and vertical opening (headline to groundgear). The headline height and vertical opening parameters were recorded using cameras (Cheng et al., 2022). A total of five rigging scenarios were tested and scaled to full-size terms, detailed in Table 1. The first rigging scenario was to connect the upper bridles to the warps at 30.5 m forward of the doors (i.e., fork connection forward of the door; Fig. 2) and tested at four flow velocities (i.e., 2.5, 2.8, 3.0, and 3.2 kt). Rigging scenarios 2 and 3 were to reduce the fork connection from 30.5 m to 20.42 and 15.24 m, respectively and tested at a flow velocity of 2.8 kt. Scenarios 4 and 5 tested the rigging scenario 3 with different warp-to-depth ratios, where the warp-to-depth ratio increased (i.e., from 2.6:1 for scenario 4 to 6.9:1 for scenario 5).

Table 1 The geometry of a scaled semi-pelagic trawl comparing rigging scenarios tested in a flume tank.

Rigging scenario	Description	Upper bridle length (m)	Tow speed (kt)	Door spread (m)	Headline opening (m)	Vertical opening (m)	Seabed clearance (m)	
1	Connect the top bridles to a 30.5 m (1.27 cm Ø) to the warp forward of door	85.4	2.5	67.8	8.0	7.2	0.8	
			2.8	67.4	7.3	5.7	1.6	
			3.0	67.1	6.9	5.0	1.9	
	3.2	67.4	6.6	4.3	2.3	
2	Connect the top bridles to a 20.42 m to the warp forward of door	75.32	2.8	66.4	7.0	6.4	0.6	
3	Connect the top bridles to a 15.24 m to a warp forward of door	70.14	2.8	66.3	6.7	6.4	0.3	
4	Warp-to-depth (ratio between the warp length and depth) 2.6:1	70.14	2.8	65.0	7.3	6.3	1.0	
5	Warp-to-depth (ratio between the warp length and depth) 6.9:1	70.14	2.8	66.4	6.3	6.3	0.0	

Figure 2 Side profile schematic of the semi-pelagic trawl system.

Field tests

Two field experiments were conducted off the west coast of Newfoundland in the GSL, CA, in April 2021 and April 2022 (Fig. 3) onboard the commercial fishing trawler F/V Lisa M (overall length 19.8 m; gross tonnage 122.5 t; engine power 700 hp; 1 hp = 746 W). The trawl used for field experiments was a groundfish balloon trawl, described in section 2.1 and Cheng et al. (2020). The trawl was spread with a pair of low-aspect trawl doors (Injector Door Limited, Søvik, Norway), which were 4 m2 in area. Fishing locations were chosen in collaboration with fishers (i.e., captain and crews) who have experience fishing redfish; fishing was carried out 24 h a day (i.e., both day and night hauls). During field experiment No. 1, the codend had to remain open because we could not land redfish due to a combination of no available redfish quota, licensing constraints, and no local market for redfish at the time of fishing. For field experiment No. 2, the codend was closed. A T90 codend (nominal 90 mm mesh size) described in Cheng et al. (2020) was used. Mesh measurements (wet) were obtained with an Omega gauge (Fonteyne, 2005), n = 60 with a mean of 89.5 mm and a standard deviation (SD) of 3.3.

Figure 3 Location of field experiments (red rectangle) in the Gulf of St. Lawrence, Canada.

Map is created using the data derived from global administrative areas (https://gadm.org/); GADM license.

Field experiment No. 1

Experimental design and data collection

Trawl mensuration included door spread, headline height, and vertical opening (headline to groundgear) using Notus trawl mensuration sensors (Notus Electronics Ltd. St. John’s, Newfoundland and Labrador, CA). Cameras and lights were used to observe the interaction between groundgear and seabed, and fish behaviour at the trawl mouth. A set up with a camera (GoPro, Woodman Laboratories, Inc., Half Moon Bay, CA, USA) and two flashlights (DIV08W diving lights from Brinyte Technology Ltd., Guangdong, China) using red light placed within waterproof housings, and connected to a plastic panel was attached to the middle of the trawl mouth. The position of the camera system was just aft of the fishing line, looking forward and at a slight angle toward the port to observe more footage of fish interacting with the groundgear and clearer documentation of when the trawl was on or off the seabed (Fig. 4). Videos collected were analyzed using Adobe Premiere Pro (Adobe Systems, Inc., San Jose, CA, USA) by a single observer. Interaction between groundgear and seabed (trawl state) was determined at the start of the tow and defined as when the trawl was fishing off- (groundgear was off the seabed) or on-seabed (groundgear was slightly off-seabed, light on seabed or hard on the seabed). Trawl state was confirmed via video. The duration of each trawl state was counted every minute (min) from underwater videos.

Figure 4 Camera placement and view at the trawl mouth.

Top left: screen capture from video collected during sea trials, groundgear is in the lower center of the shot. Top right: camera system included a camera placed in the middle and two flashlights using red light. Bottom: illustration of semi-pelagic trawl; the red triangle is the area within the center of the trawl mouth observed by a camera.

Redfish behaviour at the trawl mouth

Fish behaviour and its effects on trawl entry were analyzed based on observations and behaviours of individual redfish at the center of the trawl mouth (Figs. 4 and 5). Methods were derived from a similar behavioural study outlined by Bayse, Pol & He (2016). Variables were determined between the first detection on video until trawl entry that included trawl entrance (Entry; fish entered trawl above fishing line) or escape (Escaped; fish escaped under the fishing line) and observations and behaviours were then placed into eight categories detailed in Table 2. Fish that went out of the screen without a clear trawl entry were considered unknown and removed from analysis. Of the eight categories, fish position (Position) was considered at first detection and was split into “Above” or “Below” the fishing line. Orientation is noted by the direction of the fish head in relation to the towing direction and the middle roller of the groundgear as: the head that oriented with and against the towing direction was classified as “away” and “toward”, and the head oriented to the left and right of middle roller of the groundgear were classified as “left” and “right”, respectively. Swimming behaviour was classified into two categories, Swimming behaviour 1 and Swimming behaviour 2. Swimming behaviour 1 considered behaviours in the horizontal plane. This behaviour included swimming with (With; fish that were swimming in the direction of trawling), swimming against (Against; fish that were swimming in the opposite direction of trawling), and passive (Passive; fish that were not swimming—holding station—or lying on the seabed). Swimming behaviour 2 considered behaviours in the vertical plane. This swimming behaviour included swimming up (Up; fish that swam upward), swimming down (Down; fish that swam downward towards the seabed), and no change (NC; fish that had no changes in their swimming direction in the vertical plane).

Figure 5 Screen capture of video frames illustrates redfish behaviour at the center of trawl mouth.

Top left illustrates redfish detected at the above position, swimming against the trawl path and entering the trawl. Top right illustrates redfish detected at the above position and swimming with the trawl path and allowing the groundgear to pass below. Bottom left illustrates redfish detected below the fishing line and swimming against the trawl path and escaping under the trawl. Bottom right illustrates redfish detected at the bottom position, turning 180° before contacting the groundgear.

Table 2 Detailed description of each variable used to describe the behavioural sequence of redfish at the trawl mouth of semi-pelagic trawl.

Redfish variables	Categories	Description	
Position (A/B)	Above	Fish appear above the fishing line	
Below	Fish appear under the fishing line	
Swimming behaviour 1	Swimming with (With)	Fish swim in the same direction of the trawling	
	Swimming against (Against)	Fish swim in the opposite direction of the trawling	
Passive swimming (PS)	Fish drifted into the trawl or passed over by the groundgear	
Swimming behaviour 2	Swimming up (Up)	Fish first detected under the fishing line and rose up to enter the trawl	
	Swimming down (Down)	Fish first detected above the fishing line on the top of camera screen and swum down to escape under the footgear	
No change (NC)	Fish had no changes in their swimming direction in the vertical plane.	
Orientation	Away	Head oriented away from the trawl	
	Toward	Head oriented toward the trawl (codend)	
	Left	Head oriented to the port	
Right	Head oriented to the starboard	
Groundgear contact	Contact	Fish had contact with the groundgear	
No contact	Fish did not have contact with the groundgear	
Time	Seconds	Period between the first detection and entering the trawl or escaping under the trawl	
Trawl state	On-seabed	When the trawl is on-seabed	
Off-seabed	When the trawl is off-seabed	
Period	Day	When fishing during the day-time	
Night	When fishing during the night-time	

The variable “Contact” considered any contact between any section of the groundgear and redfish (Table 2). Other variables considered in the analysis included, time (period from first detection to trawl entry), trawl state (trawl on or off the seabed), and period (trawling during day or nighttime).

The observed effects from variables listed in Table 2 on the trawl entry of redfish were analyzed using a binomial generalized linear mixed model (GLMM) in R (version 4.2.2) statistical software (R Development Core Team, 2020). The model included trawl entry as the dependent variable, and independent variables listed in Table 2. Each individual tow (Tow) was considered a random effect on the intercept to account for variations in observations among tows due to extrinsic factors (i.e., environmental conditions, fish density, etc.). Model diagnostics were considered by investigating the data and models with the DHARMa package (Hartig, 2021) and multicollinearity with the vif function from the car package (Fox et al., 2012). Model selection was evaluated by information criterion (IC), where both the Bayesian information criterion (BIC; Schwarz, 1978) and the Akaike information criterion (AIC; Akaike, 1974) values with a correction for small sample sizes (AICc) were investigated. Initially all model combinations (n = 256) with Tow as a random effect on the intercept were run and parameters were estimated by maximum likelihood estimation using the automated model selection package glmulti (Calcagno & De Mazancourt, 2010) to select the candidate models which included important variables using IC and the relative importance of model terms plot in the glmulti package. The final models were run using the glmer function in the lme4 package (Bates et al., 2015), and the best model was determined from the minimum IC calculated from the BICtab or AICctab function from the bbmle package (Bolker, 2020). A delta IC of 2 or less indicated that models were similar, and the lowest IC was considered the best model.

Redfish behaviour under the groundgear

Redfish behaviour under the groundgear was recorded when the camera (in the same location as described above and in Fig. 4) was pointed straight down observing the area of the groundgear just under and behind the fishing line (Fig. 6). Redfish were observed before and after interacting with the passing groundgear. Noted observations and behaviours include, position (left or right), orientation (left, right, toward, away in relation to head position to the trawl path), swimming behaviour, turning (turn or no turn), trawl state (on- or off-bottom), and contact (contact or no contact). Swimming behaviour was grouped into passive swimming (PS; i.e., fish were laying on their side on the seabed, or slightly laying on the seabed with no swimming in response to upcoming trawl’ components), active swimming (i.e., swimming with or against the trawling direction; holding station), and startled reaction (swam sideways in relations to groundgear after being startled). The number of fish that contacted the groundgear was also noted for several swimming behaviour categories. Fish that passed under the groundgear (i.e., passing between rollers and rolled over by the rollers) were counted.

Figure 6 Screen capture of video frames illustrates redfish behaviour under the groundgear.

Top left and right illustrate redfish laid on their side and were pressed by a roller. Bottom left illustrates redfish was swimming against the trawl and escaped between rollers. Bottom right illustrates redfish was swimming with the tow direction and allowing the groundgear to pass over.

Field experiment No. 2

A small experiment compared both the semi-pelagic trawl described above at full-scale, as well as the trawl rigged as a conventional trawl, which simply involved the removal of the extended bridle cables and attached the warp/upper bridle back to the door. The conventional trawl setup is described in Cheng et al. (2020) and Nguyen et al. (2023). The goal of this work was to test both trawls ability to capture redfish and bycatch species. Catches were transferred from the codend to a hopper that fed a conveyer system. Redfish went directly to the fish hold and redfish total catch was estimated by the fisheries observer which is standard practice in the fishery. Bycatch was sorted, counted, and weighed to the nearest 0.1 kg by Marine Institute scientists. Large Atlantic halibut weights were visually estimated since we did not have large enough equipment to weigh them. The gear mensuration setup matched that described in Experiment 1.

The experimental fishing license of this study granted by Fisheries and Oceans Canada (NL-6020-20). The license required that all redfish catches be landed.

Results

Flume tank test and full-scale modifications

Overall, the flume tank test showed that the semi-pelagic trawl was effective at fishing off the seabed and was dynamic in fishing between off- and on- the seabed. The first scenario indicated that the seabed clearance was between 0.8 and 2.3 m when connecting the upper bridles to the warps at 30.5 m forward of the doors, and the seabed clearance increased with increasing flow velocities. Reduction in bridle extension length reduced the seabed clearance (i.e., rigging scenarios 2 and 3; Table 1). Further, increasing the warp-to-depth ratios from 2.6:1 to 6.9:1 changed the trawl from being off-seabed to on-seabed (i.e., rigging scenarios 4 and 5; Table 1). Rigging scenario 1 was adapted to the existing groundfish trawl for the subsequent field experiments by extending the upper bridle with a 38.1 m cable (1.27 cm Ø) to the warp with a G-hook. A 30.5 m cable (1.27 cm Ø) was attached between previously described connections to the door. An additional 4.27 m chain was added to the aft of the lower bridles (Fig. 2).

Field experiment No. 1

Gear handling, performance, and efficiency

A total of 28 tows were carried out during the experiment. One tow was not considered because a cable was wrapped up in the trawl, which affected the gear’s performance; thus, 27 valid hauls were used for analyses (Table 3). The mean depth of the fishing ground was 282.4 m(range: 255.7 to 308.9 m), the average haul duration was 166.1 min(range: 82 to 225 min), the towing speed was between 2.0 and 2.8 kt. The mean door spread was 65.5 m (range: 61.8 to 70.0 m), and the mean length of the warp deployed was 581.3 m (range: 548.6 to 640.1 m). Bottom water temperature was not measured due to a malfunctioning instrument, however according to Galbraith et al. (2022), the temperature was likely between 6 and 8 °C.

Table 3 Trawl system performance observed for each haul during field experiment No. 1.

Tow	Trawl state	Duration (min)	Warp length (m)	Door spread (m)	Headline height (m)	Vertical opening (m)	Seabed clearance (m)	Depth (m)	Tow speed (kt)	Warp to Depth Ratio	
1	Off	145	571.5	63.3	14.3	11.2	3.1	265.4	2.5	2.2	
2	Off	144	548.6	65.6	22.0	11.0	11.7	267.8	2.6	2.0	
3	Off	116	548.6	64.1	13.1	11.2	1.9	258.7	2.5	2.1	
4	Off	128	548.6	66.9	12.8	10.7	2.3	261.5	2.4	2.1	
5	On	144	548.6	68.6	11.5	15.8	–	255.7	2.7	2.1	
6	NA	177	548.6	–	–	–	–	–	2.2	–	
7	Off	188	548.6	–	12.9	11.9	2.4	258.6	2.3	2.1	
8	Off	175	548.6	–	16.4	11.7	2.8	257.4	2.4	2.1	
9	Off	160	548.6	62.5	–	–	–	276.5	2.4	2.0	
10	Off	182	548.6	61.9	17.5	11.0	8.2	283.6	2.5	1.9	
11	Off	225	571.5	63.1	15.6	–	–	282.7	2.3	2.0	
12	Off	168	594.4	66.4	16.5	–	–	288.6	2.6	2.1	
13	On	197	594.4	66.0	12.4	–	–	278.2	2.4	2.1	
14	Off	203	594.4	66.8	12.8	–	–	277.3	2.3	2.1	
15	On	198	548.6	64.8	13.5	–	–	277.1	2.5	2.0	
16	Off	187	548.6	61.8	14.4	12.9	1.8	304.1	2.3	1.8	
17	Off	118	594.4	63.4	13.4	11.9	1.7	298.5	2.2	2.0	
18	Off	165	640.1	68.3	12.2	12.1	1.1	308.9	2.5	2.1	
19	Off	185	640.1	65.6	12.4	12.0	1.0	302.0	2.3	2.1	
20	Off	189	640.1	63.6	14.4	11.8	2.9	300.7	2.2	2.1	
21	Off	82	594.4	66.0	13.8	12.1	1.1	289.8	2.3	2.1	
22	On	180	594.4	67.0	12.1	12.1	0.7	291.9	2.3	2.0	
23	On	182	594.4	68.9	12.2	12.3	0.8	292.5	2.5	2.0	
24	On	185	594.4	64.1	14.5	12.2	2.6	290.6	2.6	2.0	
25	Off	113	594.4	65.7	11.9	12.1	0.5	293.6	2.2	2.0	
26	Off	157	594.4	64.2	13.6	11.6	2.2	285.4	2.1	2.1	
27	On	175	640.1	70.0	11.2	11.8	0.9	289.2	2.4	2.2	
28	On	184	594.4	69.1	11.7	12.3	0.8	289.6	2.4	2.1	

There were no problems in handling the semi-pelagic trawl, and the hauling back process was similar to typical operations. The only exception being a slight delay (2-3 min) from when the G-hook passed through the warp winch from the connection point of the upper bridle extension, though this could be negated by splicing the upper bridle into the warp. The average observed seabed clearance of the semi-pelagic trawl was 2.5 m (ranged from 0.5 to 11.7 m; Table 3). For seven tows, the seabed clearance was not observed due to equipment malfunction. The trawl was able to perform effectively on or off-seabed as desired. Of 27 tows implemented during the field experiment, 19 were considered off-seabed, and eight were on-seabed. The total time observed for off-seabed and on-seabed tows was 38.4 h and 13.1 h, respectively. Off-bottom trawling was more accessible and consistent, whereas trawling on-bottom led to the trawl being either hard on the bottom, light on the bottom (groundgear barely touching the sea floor), or frequently coming off-seabed a short distance (observed from video). Additionally, the trawl was aimed to fish off-seabed when fishing with the tide; while fishing against the tide, the trawl was observed to fish on-seabed. This arrangement assessed the capacity of the trawl to fish in a “best case” scenario. To fish the trawl in the opposite way(e.g., off-bottom when against the tide) required letting out more warp or increasing the tow speed. Overall, the trawl fished effectively with regard to lifting the trawl system off the seabed, and the seabed clearance could be controlled by changes in towing speed and warp length.

Fish behaviour analysis

Fish behaviour was analyzed using 22 of 27 valid tows where video recordings were collected during the experiment, 19 tows (∼42 h) focused on redfish behaviour in the trawl mouth and three tows (∼6.5 h) with the camera pointed down towards the seabed.

Fish behaviour at the trawl mouth

A total of 2,196 redfish were observed, including 2099 individuals with a known trawl entry and 97 with an unknown trawl entry. Thus, redfish with an unknown entry outcome were removed from the analysis. Of the 2,099 redfish, 1,168 were observed to enter the trawl, and 931 escaped under the fishing line. The majority of redfish were first detected under the fishing line (73.1%), and 26.9% were observed above the fishing line of the trawl (Table 4). Redfish that were detected above the fishing line had a lower escape percentage (26.6%) than those seen on the bottom (50.9%). The most frequent swimming behaviour observed was fish that were swimming against the trawling direction (62.3%), second was swimming with the trawl (22.0%), followed by passive (15.8%). Escape rates for redfish swimming with the trawl (77.7%) were higher than those either passive or swimming with the trawl (62.2% and 28.1%, respectively) (Table 4). In relation to the vertical plane, many (60%) of redfish had no change in their swimming direction, 34% swimming upward, and 6.2% swimming downward. Redfish that were swimming downward had a higher escape rate (94.6%) than those which had no change in their swimming direction (63.9%); a few(0.8%) redfish that were swimming upward escaped under the trawl.

Table 4 Observed behaviour of redfish at the trawl mouth in relation to capture outcome (enter the trawl or escape under the groundgear).

A/B presents above/below, PS presents passive swimming, and NC presents no changes in their swimming direction in the vertical plane.

Variables	Observations	% Total	Entry	% Entry	Escape	% Escape	
Position (A/B)							
Above	564	26.9	414	73.4	150	26.6	
Below	1,535	73.1	754	49.1	781	50.9	
Swimming behaviour 1							
With	461	22	103	22.3	358	77.7	
Again	1,307	62.3	940	71.9	367	28.1	
PS	331	15.8	125	37.8	206	62.2	
Swimming behaviour 2							
Up	714	34.0	708	99.2	6	0.8	
Down	130	6.2	7	5.4	123	94.6	
NC	1,255	59.8	453	36.1	802	63.9	
Groundgear contact							
Contact	385	18.3	121	31.4	264	68.6	
No contact	1,714	81.7	1140	55.3	923	44.7	
Trawl state							
On-seabed	1,629	77.6	891	54.7	738	45.3	
Off-seabed	470	22.4	277	58.9	193	41.1	
Period							
Day	1,311	62.5	732	55.8	579	44.2	
Night	788	37.5	436	55.3	352	44.7	

The vast majority (81.7%) of redfish did not contact the groundgear, while 18.3% had contact (Table 4). The mean time of redfish that entered the trawl was 1.05 s (±0.07 SEM (standard error of the mean)) vs. 0.99 s (±0.02 SEM) for those that escaped underneath the fishing line. Many (77.6%) of redfish were observed when the trawl was on-seabed, compared to 22.4% when the trawl was off-seabed. Tables 5 and 6 showed the fish behaviour when the trawl was on- and off-seabed, respectively. Additionally, more redfish (62.5%) were observed during the day than the night (37.5%, Table 4).

Table 5 Fish behaviour when on-bottom trawling.

A/B presents above/below, PS presents passive swimming, and NC presents no changes in their swimming direction in the vertical plane.

Variables	n	% Total	Entry	% Entry	Escape	% Escape	
Position (A/B)							
Above	277	17.0	234	84.5	43	15.5	
Below	1,352	83.0	657	48.6	695	51.4	
Orientation							
Toward	356	21.9	230	64.6	126	35.4	
Away	597	36.6	363	60.8	234	39.2	
Left	348	21.4	169	48.6	179	51.4	
Right	328	20.1	129	39.3	199	60.7	
Swimming behaviour 1							
With	426	26.2	83	19.5	343	80.5	
Again	994	61.0	756	76.1	238	23.9	
PS	209	12.8	52	24.9	157	75.1	
Swimming behaviour 2							
NC	987	60.6	30	15.7	161	84.3	
Up	614	37.7	608	99.0	6	1.0	
Down	28	1.7	0	0.0	28	100.0	
Groundgear contact							
Yes	354	16.9	111	31.4	243	68.6	
No	1,275	60.7	780	61.2	495	38.8	
Period							
Day	1,026	63.0	564	55.0	462	45.5	
Night	603	37.0	327	54.2	276	45.8	

Table 6 Fish behaviour when off-bottom trawling.

A/B presents above/below, PS presents passive swimming, and NC presents no changes in their swimming direction in the vertical plane.

Variables	n	% Total	Entry	% Entry	Escape	% Escape	
Position (A/B)							
Above	287	61.1	180	62.7	107	37.3	
Below	183	38.9	97	53.0	86	47.0	
Orientation							
Toward	215	49.5	131	52.2	120	47.8	
Away	120	27.6	89	74.2	31	25.8	
Left	28	6.5	20	71.4	8	28.6	
Right	71	16.4	37	52.1	34	47.9	
Swimming behaviour 1							
With	35	7.4	20	57.1	15	42.9	
Against	313	66.6	184	58.8	129	41.2	
PS	122	26.0	52	24.9	157	75.1	
Swimming behaviour 2							
NC	268	57.0	170	63.4	98	36.6	
Up	100	21.3	100	100.0	0	0.0	
Down	102	21.7	7	6.9	95	93.1	
Groundgear contact							
Yes	31	6.6	10	32.3	21	67.7	
No	439	93.4	267	60.8	172	39.2	
Period							
Day	324	70.4	189	58.3	135	41.7	
Night	146	29.6	88	60.3	58	39.7	

The automated model selection process showed convergence issues which were improved by removing the variable orientation. Thus, the total number of models ran was 128. AICc had nine models within ∼2 delta AICc and BIC had only two. AICc had lower IC values for more complicated models in comparison to BIC. The relative importance plots were similar between the different ICs. Both considered the three most important variables to be position, swimming behaviour 1, and Swimming behaviour 2. However, AICc valued swimming behaviour 1 equal to position and swimming behaviour 2 where BIC had swimming behaviour 1 at ∼20% lower than the other two variables. Thus, following the principal of parsimony and a clear model preference, BIC was used to determine that the best model included position and swimming behaviour 2 (Table 7).

Table 7 BICc values were estimated for two best models.

Model	BIC	dBIC	df	weight	
Trawl entry ∼Position + Swimming behaviour 2 + (1 — Tow)	782.6	0.0	5	0.77	
Trawl entry ∼Position + Swimming behaviour 1 + Swimming behaviour 2 + (1 — Tow)	785.0	2.4	7	0.23	

Redfish behaviour under the groundgear

A total of 603 redfish were observed under the trawl, just behind the fishing line. Of those, 528 clearly escaped, while 75 had an unknown escape (swam out of view; though likely escaped) and were not further described. In general, redfish escape behaviour under the groundgear was observed in three ways: 291 (55%) were observed to show passive swimming (including 96 individuals laid on the seabed and 195 individuals were sitting or touching the seabed without swimming), 196 (37%) swam actively (i.e., swimming with or against the trawling direction), and 41 (7.7%) showed a startled response and swam sideways in relation to towing direction (Table 8). These different swimming behaviours led to differences in the ways that fish passed under the groundgear, where a total of 461 redfish passed between groundgear rollers versus 67 that were rolled over by the groundgear rollers. Most fish that swam against the upcoming trawl escaped through the escape opening between rollers, and these fish did not contact the groundgear. Fish that swam with the trawl direction kept the upcoming trawl at a short distance and mostly returned to escape between rollers until the trawl came closer. Though occasionally some of these fish were passed over or rolled over by the rollers as they swam slower than the upcoming trawl. Redfish that were startled in a sideways direction were observed to contact the front side or impinge with the inside of the rollers as they passed under the trawl. Most fish that have contact with the groundgear was observed when trawling on the seabed.

Table 8 Observed behaviour of escaping redfish at the trawl mouth.

L/R stands for left/right, PS presents passive swimming.

Variables	n	% Total	Contact	% Contact	No contact	% No contact	
Position (L/R)							
Left	336	63.2	104	31.0	232	69.0	
Right	196	36.8	71	37.0	121	63.0	
Orientation							
Left	77	14.5	40	52.0	37	48.0	
Right	125	23.7	51	40.8	74	59.2	
Toward	142	26.9	27	19.0	115	81.0	
Away	184	34.8	57	31.0	127	69.0	
Swimming behaviour							
Against	79	15.0	39	49.4	40	50.6	
With	117	22.2	65	55.6	52	44.4	
Startle	41	7.7	28	68.3	13	31.7	
PS	291	55.1	43	14.8	248	85.2	
Turning							
Turn	193	36.6	114	59.1	79	40.9	
No turn	335	63.4	61	18.2	274	81.8	
Trawl state							
On-seabed	294	55.7	168	57.1	126	42.9	
Off-seabed	234	44.3	7	3.0	227	97.0	
Contact							
Contact	175	33.1	113	64.6	62	35.4	
No contact	353	66.9	348	98.6	5	1.4	

Field experiment No. 2

A total of 15 tows were completed during the experiment, including six hauls for the experimental trawl and nine hauls for the conventional trawl. Due to circumstances that included poor weather, vessel breakdowns, gear mensuration malfunctions, and lower than expected catch at the beginning of trials; comparative fishing was not attempted. Thus sampled hauls could not be paired for comparison and any day and nights effects could not be delineated. Collected data, including gear mensuration and catch of redfish and bycatch species were used for preliminary assessment of each trawl’s performance, ability to catch redfish off the seabed and potentially avoiding bycatch species in the catch regarding changing the warp length. Two different warp length ranges were conducted to improve redfish catch rates, including shorter warp length during first ten hauls (758.5 m mean in length, range 731.5–789.9 m) and longer warp length during last five hauls (849.3 m mean, range 823.0–890.6 m).

For the trawl’s performance, seabed clearance was only measured for the final two tows due to an equipment malfunction. For these tows, the trawl averaged 1.0 m (SD = 0.2) off of the seabed. The mean warp length was 791.0 m (range: 731.5 to 890.6 m). For the shorter warp length range, the average haul duration was 93.2 min (range: 55 to 148 min), the mean tow speed was 2.5 kt (range: 2.3 to 2.7 kt), the mean door spread was 61.1 m (range: 58.7 to 62.8 m), the mean depth of the fishing ground was 341.0 m (range: 330 to 354.7 m), and the average RPM(engine revolution per minute) was 1,405.6 (range: 1,356 to 1,459). For the longer warp length range, the average haul duration was 91.6 min (range: 55 to 125 min), the mean tow speed was 2.4 kt (range: 2.3 to 2.5 kt), the mean door spread was 60.7 m (range: 57.4 to 65.7), the mean depth of the fishing ground was 350.7 m (range: 334.6 to 361.1 m), and the RPM was 1441.8 on average (range: 1,419.5 to 1,464). The mean bottom water temperature was 7.3 °C (SD = 0.1).

For catches of redfish, a total of 26,341.0 kg (15 tows) was estimated to be caught during the experiment. The total catch of redfish caught during the shorter warp length tows estimated for the experimental trawl and the conventional trawl was 1,129.5 kg and 4,866.2 kg, respectively. The total catch of redfish caught during the longer warp length tows was 2,218.1 kg estimated for the experimental trawl and 18,143.7 kg estimated for the conventional trawl.

For bycatch species, a total of 15 bycatch species were observed during the experiment. Of those bycatch species, four species had capture totals above 50 kg, including Atlantic halibut (340.0 kg conventional trawl; 13.0 kg semi-pelagic trawl), white hake (175.2 kg conventional trawl; 13.0 kg semi-pelagic trawl), Atlantic cod (110.5 kg conventional trawl; 8.0 kg semi-pelagic trawl), and thorny skate (Amblyraja radiata; 75.0 kg conventional trawl; 3.0 kg semi-pelagic trawl). During the short warp length tows, the total catch of Atlantic halibut, white hake, Atlantic cod, and thorny skate were 31, 68.2, 40.5, and 17 kg, respectively. The total capture of these bycatch species during the long warp length tows (i.e., 322 kg, 132 kg, 78 kg, and 61 kg estimated for Atlantic halibut, white hake, Atlantic cod, and thorny skate, respectively).

Discussion

This study developed a functional semi-pelagic trawl via simple modifications added to the warps and bridles of a typical groundfish trawl. The flume tank and field experiments provided evidence for the engineering effectiveness of the rigging to lift the trawl from the seabed while maintaining the doors on bottom, providing potential for reduced seabed impacts and bycatch reduction. The operation of the trawl, including handling and fishing, were very similar to the conventional trawl and a few more modifications could make this fishing process almost identical (i.e., splicing the bridles to the warps). The fishing experiment, though small and preliminary, shows promise for this gear as commercial catch rates of redfish were observed and some bycatch species may be captured at reduced rates, though catch rates were not compared in this study. Further work is required to fully understand how best to use the gear. Importantly, semi-pelagic trawling for redfish could provide a reasonable solution for fishers switching from targeting shrimp to the future redfish fishery in the GSL. Such a change reduces the economic impact of transitioning from a bottom trawl to a pelagic/midwater trawl.

How groundfish respond to fishing gear can influence their trawl entry, as shown in previous studies (Kim & Wardle, 2003; He, Smith & Bouchard, 2008; Winger, Eayrs & Glass, 2010; Brinkhof et al., 2017). Fish responses at the trawl mouth can be related to their vertical distribution, reaction tendency, and swimming behaviour, which can lead fish to enter or escape the trawl different ways (Main & Sangster, 1981; Godø& Walsh, 1992; Kim & Wardle, 2003). Similarly, redfish were observed in the current study distributed at different heights at the center of the groundgear, separating into different positions above (73.1%) and under the fishing line (26.9%). These different heights altered the escape rates of redfish. Our study estimated a little over half of redfish observed on the bottom escaped, where most redfish observed above the fishing line entered the trawl. This implies that as a poor swimmer, redfish may seek escape openings under the groundgear for escaping rather than rising above the fishing line, and becoming available for capture. Of note, 26.6% of redfish observed above the fishing line escaped. These escapees may be explained by the escape behaviour that redfish were observed at heights close to the fishing line, particularly small fish, likely searching for escape openings under the fishing line (Engås, Jacobsen & Soldal, 1988).

Our analysis showed that the swimming behaviour of redfish directly influences the trawl entry. In terms of swimming direction in relation to tow direction, many redfish that were swimming against the oncoming trawl tend to enter the trawl; these tended to enter the trawl rather than escape under the fishing line. These fish were observed to maintain their swimming direction from the first direction to entering the trawl. This behaviour is similar to the optomotor response of haddock near the groundgear, described by Kim & Wardle (2003). Contrary, most redfish that were swimming in the towing direction (i.e., swimming with) and passively swimming escaped under the fishing line. Many of these fish were observed to swim less than the tow speed, allowing the trawl to pass over; some showed erratic swimming and therefore escaping.

In addition to swimming behaviour, a large proportion of redfish was swimming up above the fishing line, resulting in most fish being caught. This swimming behavior is similar to Main & Sangster (1981) and Godø& Walsh (1992), who found that roundfish remained close to the bottom but rose above the fishing line when they came into contact with the footgear. Inversely, a proportion of fish was observed to swim down, leading to the highest escape rate relative to other swimming behaviours. These fish were observed just a bit above the fishing line, oriented toward the approaching trawl, and tended to swim down in order to seek the spaces under the footgear for escape.

Previous observations revealed that contacting the groundgear can lead to fish and organisms in relation to the seabed to become available for capture (Nguyen et al., 2014; Bayse, Pol & He, 2016). However, this study found over half of the individual redfish was observed to enter the trawl without groundgear contact even when the trawl was on or off the seabed (Tables 5 and 6). These fish were first detected on the bottom and rose above the fishing line when approaching the upcoming trawl’s groundgear as discussed above. This behaviour is similar to the entry behaviour of roundfish, observed by Main & Sangster (1981) and Thomsen (1993). There was a considerable proportion of redfish that had contact with the groundgear and escaped through spaces between rollers. These escaping fish may erratically respond to the groundgear in a short distance, suddenly dart away by using a kick and glide gait, or run over by the groundgear for escape (Kim & Wardle, 2003).

The fate of escapees from capture potentially influence populations (Main & Sangster, 1983; Ingólfsson & Jørgensen, 2006; Ingólfsson et al., 2007). Observations have shown that the survival capacity of escaping fish under the trawl was related to their groundgear contact (Ingólfsson & Jørgensen, 2006; Nguyen et al., 2014; Bayse, Pol & He, 2016). The authors suggested that contacting the groundgear induced more injured fish than did no contact, potentially resulting in the mortality of fish when escaping under the fishing line. In the current study when the trawl was off the seabed, redfish that escaped under the fishing line typically did not have contact with the groundgear. While redfish mortality by escaping under the trawl was not quantified, reducing groundgear contact by using a semi-pelagic trawl might reduce injuries, physiological stress, and potential predation risk, leading to a reduction in redfish mortality. This would be beneficial for upcoming redfish fisheries in the GSL with regard to avoiding declines in population due to escape mortality.

The second field experiment provided a first-look at data of catches of redfish for a trawl on and off bottom. These results are preliminary and should be interpreted with caution. Only 15 tows were tested and we were unable to have any sort of balanced design (different warp-to-depth ratios, day vs. night, alternating of treatments). Catch rates were lower than expected for the first 10 tows, and after four tows the trawl was adjusted to fish on the bottom, entirely removing the semi-pelagic rigging. An increase in warp (and warp-to-depth ratio) directly led to catch rates at least doubling. Unfortunately, only five tows were fished like this, and only two with the semi-pelagic setup. At this point, the vessel broke down and ended further investigation. This is not enough to directly compare these gears by any means, however, it does allow a preliminary proof of concept that the semi-pelagic trawl is at least capable of capturing commercial quantities of redfish (Tow 15). What drove the improved catch rates is not specifically known; unfortunately, the gear mensuration equipment was not functioning for the first 13 tows only allowing an investigation into the last tows. Likely, the increase in warp-to-depth ratio increased the trawl opening horizontally while reducing the headline height, allowing for a more efficient trawling scenario. However, the collected data do not necessarily point to this conclusion, which is difficult when only evaluating two tows, but this assumption is based on experience and the literature (Fujimori et al., 2005).

The behavioural differences of fish at the trawl mouth have been utilized to modify the gear regarding increasing vertical distance between the fishing line and seabed to separate flatfish and other species related to the seabed from the catch (Main & Sangster, 1985; Engås, Jørgensen & West, 1998; Krag et al., 2010). Flatfish response to an oncoming trawl has been observed either resting close to the seabed and passed over by the groundgear (Walsh & Hickey, 1993) or rising to enter the trawl within 1 m from the seabed (Main & Sangster, 1981; Bublitz, 1996; Underwood et al., 2015). Thus, it is not surprising that there were few captures of flatfish and skate species when the trawl was off the seabed at 1.0 m on average. The semi-pelagic trawl relative to the conventional trawl showed a trend to avoid the capture of Atlantic cod and white hake in the catch, thought no statistical comparison was made. Even though this is a preliminary result, this could be explained by the behavioural tendency of fish at the trawl mouth. When aggregating at the center of the trawl mouth, large individuals with greater swimming capacity tend to rise from the seabed and enter the trawl (Main & Sangster, 1981; Thomsen, 1993; Ingólfsson & Jørgensen, 2006; Krag et al., 2010). Some of these fish may reach the heights available to be captured by the semi-pelagic trawl.

Conclusions

A semi-pelagic trawl was evaluated for use in the GSL redfish fishery to target redfish off the seabed. This approach shows promise as fisher’s transition from targeting Northern shrimp to redfish. Here, we have documented steps to take to modify a traditional groundfish trawl to a semi-pelagic trawl. Potential unintended benefits of using this gear include reducing negative seabed impacts. Doors would still cause damage, however having the groundgear off the seabed would greatly reduce the bottom impact when compared to a traditional trawl. Logically, demersal bycatch could be reduced since the gear is off of the seabed. Results here are too preliminary to have much confidence, however these results provide important proof to support results from Fisheries and Oceans Canada (2020) that the reduction of bycatch species, such as Atlantic halibut, white hake, and Atlantic cod could be advanced by applying the semi-pelagic trawl. Pelagic trawling traditionally was most successful when redfish were highly aggregated (Duplisea, 2018). Video analyses here, though only describing a small portion of the trawl mouth area, suggest that redfish can avoid capture by going under the trawl, and perhaps at times of day/season semi-pelagic trawling could be more or less effective than bottom trawling. Future research should investigate this further. Importantly, here we show that this fishing technique can capture a commercial quantity of redfish, which was indicated as a necessary step in gear innovation by the Ocean Frontier Institute’s Northwest Atlantic Redfish Symposium (Cadigan et al., 2022).

Supplemental Information

Data S1 Supplemental data

Observation data, which observed redfish behaviour at the trawl mouth and redfish behaviour under the groundgear during the first experiment, redfish data, which was weighted during the second experiment, bycatch data, which was weighted for bycatch capture by the trawl during the experiment, and supplemental workup, which provides the R codes to analyze the redfish behaviour at the trawl mouth.

Click here for additional data file.

We would like to thank Captains Rodney and Les Dredge and the crew of the F/V Lisa M for their assistance. Additionally, we would like to thank Mark Santos (Fisheries and Marine Institute), Francis Parrot (Notus Electronics), Morgan Snook (Hampidjan Canada Ltd.), and David Kelly (Hampidjan Canada Ltd.) for their kind help during the project.

Additional Information and Declarations

Competing Interests

Author Contributions

Animal Ethics

Data Availability

The authors declare there are no competing interests.

Vang Y. Nguyen performed the experiments, analyzed the data, prepared figures and/or tables, authored or reviewed drafts of the article, and approved the final draft.

Shannon M. Bayse conceived and designed the experiments, performed the experiments, analyzed the data, prepared figures and/or tables, authored or reviewed drafts of the article, and approved the final draft.

Paul D. Winger conceived and designed the experiments, performed the experiments, authored or reviewed drafts of the article, and approved the final draft.

Harold DeLouche conceived and designed the experiments, performed the experiments, authored or reviewed drafts of the article, and approved the final draft.

George Legge conceived and designed the experiments, performed the experiments, authored or reviewed drafts of the article, and approved the final draft.

The following information was supplied relating to ethical approvals (i.e., approving body and any reference numbers):

The experimental fishing license granted by Fisheries and Oceans Canada. The license required that all redfish catches be landed.

The following information was supplied regarding data availability:

The raw measurements are available in the Supplementary File.

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
