# Peer review of "Developing a semi-pelagic trawl to capture redfish in the Gulf of St. Lawrence, Canada"

_PeerJ, doi:10.7717/peerj.16244_

## Round 0.1 · original submission · Major Revisions

Dear Authors

The reviewers have commented on your manuscript. You can find the attached reports. Based on the comments and suggestions of the expert reviewers, unfortunately, a major revision is needed for your article.

I would like to request that you check and correct the manuscript step by step based on the reports.

Reviewer 1 ·

Basic reporting

I consider the basic reporting in the manuscript as required by this journal as sufficient. Specifically:

1. The manuscript is written in a clear and professional language; however, I found few language errors and some phrases that could, in my opinion, benefit from re-wording. I have mentioned them below this comment. Since I'm not a native English speaker, another round of language correction could potentially further improve the language in the manuscript.
The abstract is relatively long and detailed. Maybe consider removing some parts that belong more to the materials and methods?

2. The literature references mentioned in the manuscript I consider relevant and sufficient to provide the background for this study. This can be further improved to enable the reader to more understand the fishery you are describing (with potentially using the same references but with more background explanation?). Specifically:
Line 55: what do you mean by "high proportion" and "small fish"? Were there any previous MLS regulations?
Line 61 ("groundfish trawl currently used to target redfish"): since I'm not familiar with this fishery, this confused me because elsewhere in the manuscript you mention that the fishery is closed since the 90s and about to be opened (but not so far?). Please clarify this.
Lines 75-89: Please mention whether these bycatch species are commercially exploited if captured or do this bycatch need to be avoided?
Line 84: What is Unit 1?
Line 87 ("10.5%"): Unclear. Of the total catch? In weight?
Lines 87-89: Unclear to me. And again, which species constitute a concern and which ones are commercial? MLS?
Lines 107-110: When was this technique used? Was it before the moratorium? You could also explain more here (with references) what are the limitations (if any) with this system and how (if) this should be further developed and more regarding how this is linked with the present study. More emphasis on why are these modifications suggested in the manuscript relevant.

Maybe add a reference for sentence in lines 236-237 if you can.


3. I find the structure of the article in general sufficient. Please add section numbers throughout the manuscript since you are referring to sections in the manuscript text (i.e., line 167). Figures are relevant to the manuscript.
I could suggest adding a schematic drawing in the introduction close to lines 102-104, showing the semi-pelagic trawl rigging used in the fishery earlier. However, please consider this just as a suggestion.
Figure 6 is a bit unclear when presented as image. Have you considered adding small videos instead if it is allowed by the journal requirements?
Tables are sufficient.

4. In the manuscript introduction, I suggest rephrasing text in lines 125-131. In the current manuscript version, this text sounds more like materials and methods to me. Please consider rephrasing as clear aims for this research or particular research questions. Especially this is important to understand which of the mentioned aspects are assessed by the study (for example, the study does not assess size selectivity and species compositions even though it is mentioned in the introduction). In my opinion, at the end of the introduction, it should clearly state what this research is focusing on (i.e., assessing possible modifications in flume tank and assessing escape/entry of redfish in the field). Furthermore, I have additional comments and concerns regarding Experiment two and associated aims and results which I address under point "Validity of the findings".



Some small language and style suggestions that I noticed while reviewing the manuscript:
Line 54: "long-living species"
Line 54: Consider "can be susceptible" instead of "are susceptible"
Line 61: Use "size selectivity to reduce" instead of "size selectivity - reduce"
Line 61: Use "undersized fish in" instead of "undersized fish - ". Further, please see my earlier comment about this sentence.
Line 66: Maybe "where" instead of "whereby"?
Line 70: "Diamond mesh codend" instead of "diamond codend".
Line 71: You can as well remove the part of the sentence after the reference to Pol (2016). It is a bit repetitive as written.
Line 73: Consider "potential bycatch of undersized redfish".
Line 74: Break the long sentence and start a new on by changing "which" to "This".
Line 75-76: "..with many other groundfish species..".
Lines 77-79: This sentence feels like repetition.
Line 83: Change "overlap" to "overlaps".
Line 123: "Further" develop?
Lines 169-170: "fisher's experience" and "24 h a day" - unclear, consider rewording and maybe adding more details.
Line 201 "went off screen": Please change the wording.
Lines 207- 215: Couple of long sentences, suggest splitting them up. Further, suggest rewording in line 212.
Line 227: add "where" after "..criterion (IC), ".
Line 228: "the" before Akaike?
Line 229: "values" instead of "value".
Line 231: maximum likelihood estimation.
Line 232 "narrow down": Consider changing the wording.
Line 246 and elsewhere where you mention "sitting or touching the seabed": Consider change of the wording.
Line 251: Change "was" to "where".
Line 258: Remove "or avoid". The meaning of the sentence is clear. Further, add "species" after "bycatch".
Line 289: Change "unable to be" to "not".
Line 310: Change to "tows where video recordings were collected".

Experimental design

The topic of this manuscript is a primary research and within the aims and scope of the journal in my opinion. The aim of the manuscript is mentioned in the introduction; however, as suggested I would consider changing the last part of the introduction by adding clear aims of the research or clear research questions.
In the introduction, more emphasis should be put on the species selectivity instead of size-selectivity (lines 60-72) since, as I understand it, the aim of the modifications is to limit bycatch of other species? If not, this should also be more explained how the proposed modifications could affect this.
Please add that the statistical analysis were done using R (version ....) somewhere in materials and methods (i.e., line 221?).

The methods on how the trials were conducted are described in detail, also including the limitations during the study, especially for the Experiment 2. Please see my comments regarding this in the next section.

For Experiment 1, I have one concern which relates to the wording. In the results, you often mention "capture outcome". Since the codend was open during Experiment 1, use of this term can be misleading for the reader since it does not take into consideration the further selection process. The fact that the codend was open is well-described; however, I would suggest that you rephrase the words "capture outcome" to "entry outcome" or similar which, in my opinion, would be more correct and clear in this case.
Also, regarding Experiment 1, I suggest removing "catch efficiency" in line 39 of the Abstract. This is not assessed in the experiment as I understand it.

Validity of the findings

My main concern in this study is related to Experiment 2 and the conclusions that are mentioned in the results and discussion section of the manuscript.
The authors mention the limitations regarding the data collection during this part of the trials. Authors should consider whether these conclusions can be made from the results obtained that would actually allow making any comparisons of both setups. Specifically, the authors mention that due to the challenges comparative fishing was not attempted. Therefore, I think that this part should either be removed or it needs more of a carefully description if and to what extent these data can be used to any preliminary assessments of how each trawl performed.

·

Basic reporting

This manuscript investigates to development of a semi-pelagic trawl to target redfish (Sebastess spp.) and potentially reduce the capture of bycatch species and seabed impacts in the Gulf of St. Lawrance, Canada. The title and abstracts accurately reflect the content of the manuscript and provide a concise summary of the study's objectives, methods, and findings. The chosen keywords appropriately represent the subject matter of the research. The objectives stated in the manuscript are clear and appropriate, considering the focus on developing a semi-pelagic trawl to target redfish and reduce bycatch species and seabed impacts in the Gulf of St. Lawrence, Canada.

Experimental design

The research topic falls within the scope of the journal. The manuscript provides informative descriptions of the materials and methods employed in the study. The analysis of the data is well-defined, indicating a sound experimental design.

Validity of the findings

The manuscript acknowledges the utilization of previous knowledge, specifically the use of French rigging, to address the issue of reducing bycatch species and seabed impacts. This demonstrates the integration of existing techniques with the proposed semi-pelagic trawl, enhancing the validity of the findings.

Additional comments

I found the manuscript interesting and believe it would be useful for readers across multiple countries. Therefore, I suggest accepting the manuscript.

---

## Round 0.2 · accepted · Accept

I evaluated the revised version of your manuscript. I would like to thank you for considering all reviewer comments and suggestions. I am pleased to inform you that your article has been accepted
Sincerely yours

Reviewer 1 ·

Basic reporting

The present version of the manuscript is clear, supported by relevant background information and references. The structure corresponds to a scientific publication, and the figures and tables in the current manuscript version are sufficient. The reported results are relevant to the aim of this study.
In my opinion, the manuscript is well improved after the first round of revisions, and I thank the authors for considering my previous comments.

Experimental design

The aims of this study are well defined and is within the aims and scope of the journal. Methods are described in sufficient detail.

Validity of the findings

The revised version of the manuscript describes the conclusions well and also highlights the limitations of the study.